# Recent Progress in Electrocatalytic Reduction of CO$_2$

Chaojun Ren [1], Wei Ni [1] and Hongda Li [2,*]

1 Beijing Aerospace Propulsion Institute, No.1 South Dahongmen Road, Beijing 100076, China
2 Liuzhou Key Laboratory for New Energy Vehicle Power Lithium Battery, School of Electronic Engineering, Guangxi University of Science and Technology, Liuzhou 545006, China
* Correspondence: hdli@gxust.edu.cn; Tel.: +86-152-0144-0207

**Abstract:** A stable life support system in the spacecraft can greatly promote long-duration, far-distance, and multicrew manned space flight. Therefore, controlling the concentration of CO$_2$ in the spacecraft is the main task in the regeneration system. The electrocatalytic CO$_2$ reduction can effectively treat the CO$_2$ generated by human metabolism. This technology has potential application value and good development prospect in the utilization of CO$_2$ in the space station. In this paper, recent research progress for the electrocatalytic reduction of CO$_2$ was reviewed. Although numerous promising accomplishments have been achieved in this field, substantial advances in electrocatalyst, electrolyte, and reactor design are yet needed for CO$_2$ utilization via an electrochemical conversion route. Here, we summarize the related works in the fields to address the challenge technology that can help to promote the electrocatalytic CO$_2$ reduction. Finally, we present the prospective opinions in the areas of the electrocatalytic CO$_2$ reduction, especially for the space station and spacecraft life support system.

**Keywords:** carbon dioxide; electrocatalytic reduction; electrocatalyst; electrolyte; reactor design; spacecraft life support system

## 1. Introduction

With the continuous development of human science and technology, space exploration will face unprecedented opportunities and challenges [1–5]. Currently, realizing the recycle of materials in closed space stations is the best solution to solve the problem of water (H$_2$O) and oxygen (O$_2$) resupply faced by human activities, which not only can extend the duration of manned missions and reduce transportation costs, but also is a prerequisite for interstellar migration [6–10]. As the end product of human respiration and metabolic processes, carbon dioxide (CO$_2$) carries a large amount of oxygen [11–13]. Therefore, recovering O$_2$ from waste CO$_2$ is an important way to achieve resource recycling and reduce transportation costs for medium- and long-term manned extraterrestrial missions [14–18].

The concentration of CO$_2$ in the space station directly affects the life and health system of the astronauts [19]. To reduce the concentration of CO$_2$ (<0.5%) and maintain the balance of air components, it is necessary to treat the CO$_2$ emissions at 0.7–1.0 kg per person per day. How to better transform and recycle CO$_2$ in the space station has become the focus of researchers [20–23]. The methods of CO$_2$ transformation can be classified into two categories: physical absorption and chemical conversion [24,25]. The chemical conversion mainly includes thermochemistry, photocatalytic reduction, electrocatalytic reduction, and photoelectrocatalytic reduction [26–29]. Chemical conversion can capture and immobilize CO$_2$ or convert it into useful low-carbon fuels, such as CO, CH$_4$, HCOOH, CH$_3$OH, etc. [30–34].

In order to satisfy the requirements of future manned space missions, the development of a new generation of CO$_2$ conversion and oxygen production technologies has become the focus of current space technology research and development [35–39]. Currently, the CO$_2$ reduction techniques used in space stations include Sabatier, Bosch, CO$_2$ electrolysis,

CO$_2$ pyrolysis, and other reduction methods [40–44]. However, the direct conversion of CO$_2$ to small organic molecules is more attractive, and this method can yield more valuable products [45–47]. Among the numerous CO$_2$ conversion methods, electrochemical reduction of CO$_2$ has the advantages of atmospheric temperature, normal pressure, low energy consumption, and minimal environmental pollution [48–51]. Besides, the electrochemical reduction of CO$_2$ can effectively overcome the higher redox potential of reaction intermediates, which has better application prospects and significance [52–54]. However, the electrochemical conversion of CO$_2$ still faces many challenges, and it needs to satisfy two basic criteria, i.e., high energy efficiency and high reaction rate [55–59].

Here, we summarize the related works in the fields to address the challenge technology that could help to promote the electrocatalytic CO$_2$ reduction. Finally, we present the prospective opinions in the areas of the electrocatalytic CO$_2$ reduction, especially for the space station and spacecraft life support system.

## 2. The Mechanism of Electrocatalytic Reduction of CO$_2$

The electrocatalytic reduction of CO$_2$ often occurs at the interface between the electrode and the electrolyte, and the electrolyte is usually an aqueous solution of potassium bicarbonate [19]. As shown in Figure 1, the multiphase catalytic process generally consists of four main steps: (i) The CO$_2$ in the solution diffuses to the surface of the working electrode. (ii) The electrocatalyst adsorbs CO$_2$ from the solution. (iii) Electron transfer or proton migration is used to cleave C-O bonds or form C-H bonds. (iv) The generated products are detached from the electrocatalyst surface and diffused into the electrolyte [60,61]. The adsorption of CO$_2$ on the electrocatalyst surface is an important step in the whole reaction. During the adsorption process, the C=O bonding is strongly perturbed by the substrate, and the electrons are shared between the CO$_2$ and the catalyst. A number of reaction mechanisms have been proposed for the conversion of CO$_2$. It is widely accepted that the reactive species are the neutral hydrated CO$_2$ molecules, and they will convert into CO$_2$$^{\bullet-}$ in the adsorption process on most of the main-group metal electrodes in aqueous media. Then the absorbed CO$_2$$^{\bullet-}$ reacts with the H$_2$O molecules to form HCO$_2$$^{\bullet}$. The intermediate would convert into to HCO$_2$$^-$ because of its unstable unpaired electron, followed by the desorption of HCO$_2$$^-$ species [60].

**Figure 1.** Reaction mechanism of electrochemical CO$_2$ reduction on electrodes in aqueous solutions and the formation paths for the five main C1 products during the reduction process. Reprinted from Ref. [60], copyright (2017), with permission from Wiley.

Because of the complexity of electrocatalysis, this leads to the fact that the chosen electrocatalyst and the applied electrode potential have a strong influence on the final reduction product [62]. In general, the reaction products are carbon compounds with

different oxidation states [63,64]. The electrocatalytic reduction mechanism of $CO_2$ is a complicated process that involves multiple electron transfer reactions, typically including 2-, 4-, 6-, or 8-electron reaction pathways [65,66]. Table 1 summarizes the corresponding standard reduction potentials for the C1 products (C, CO, HCOOH, $CH_2O$, $CH_3OH$, and $CH_4$) and C2 products (C, CO, HCOOH, $CH_2O$, $CH_3OH$, and $CH_4$) obtained by electrocatalytic reduction of $CO_2$. Overall, the required potential to generate C=O products (except $H_2C_2O_4$) is greater than that of generating compounds containing C–H and C–OH.

**Table 1.** The standard potential for the conversion of $CO_2$ to various C1 and C2 products in aqueous solution under standard conditions (1.0 atm and 25 °C). Reprinted from Ref. [45], copyright (2014), with permission from the Royal Society of Chemistry.

| Half Electrochemical Thermodynamic Reactions | Standard Potentials (V vs. SHE) |
|:---:|:---:|
| $CO_2$ (g) + $2H^+$ + $2e^-$ = HCOOH (l) | −0.250 |
| $CO_2$ (g) + $2H_2O$ (l) + $2e^-$ = $HCOO^-$ (aq) + $OH^-$ | −1.078 |
| $CO_2$ (g) + $2H^+$ + $2e^-$ = CO (g) + $H_2O$ (l) | −0.106 |
| $CO_2$ (g) + $2H_2O$ (l) + $2e^-$ = CO (g) + $2OH^-$ | −0.934 |
| $2CO_2$ (g) + $2H^+$ + $2e^-$ = $H_2C_2O_4$ (aq) | −0.500 |
| $2CO_2$ (g) + $2e^-$ = $C_2O_4{}^{2-}$ (aq) | −0.590 |
| $CO_2$ (g) + $4H^+$ + $4e^-$ = C (s) + $2H_2O$ (l) | 0.210 |
| $CO_2$ (g) + $2H_2O$ (l) + $4e^-$ = C (s) + $4OH^-$ | −0.627 |
| $CO_2$ (g) + $4H^+$ + $4e^-$ = $CH_2O$ (l) + $H_2O$ (l) | −0.070 |
| $CO_2$ (g) + $3H_2O$ (l) + $4e^-$ = $CH_2O$ (l) + $4OH^-$ | −0.898 |
| $CO_2$ (g) + $6H^+$ + $6e^-$ = $CH_3OH$ (l) + $H_2O$ (l) | 0.016 |
| $CO_2$ (g) + $5H_2O$ (l) + $6e^-$ = $CH_3OH$ (l) + $6OH^-$ | −0.812 |
| $CO_2$ (g) + $8H^+$ + $8e^-$ = $CH_4$ (g) + $2H_2O$ (l) | 0.169 |
| $CO_2$ (g) + $6H_2O$ (l) + $8e^-$ = $CH_4$ (g) + $8OH^-$ | −0.659 |
| $2CO_2$ (g) + $12H^+$ + $12e^-$ = $CH_2CH_2$ (g) + $4H_2O$ (l) | 0.064 |
| $2CO_2$ (g) + $8H_2O$ (l) + $12e^-$ = $CH_2CH_2$ (g) + $12OH^-$ | −0.764 |
| $2CO_2$ (g) + $12H^+$ + $12e^-$ = $CH_3CH_2OH$ (l) + $3H_2O$ (l) | 0.084 |
| $2CO_2$ (g) + $9H_2O$ (l) + $12e^-$ = $CH_3CH_2OH$ (l) + $12OH^-$ | −0.744 |

Thermodynamically, the equilibrium potential of $CO_2$ reduction is comparable to the equilibrium potential of hydrogen evolution reaction (HER) [67,68]. This means that the electrocatalytic reduction of $CO_2$ is accompanied by severe HER, which leads to a decrease in the efficiency of the $CO_2$ reduction. In addition, the $CO_2$ reduction products possess small thermodynamic potential difference, indicating that it is difficult to reduce $CO_2$ to specific products with good selectivity and conversion efficiency in the current $CO_2$ reaction [69,70]. Driving the $CO_2$ reduction reaction often requires a larger overpotential, which further contributes to the technical difficulty of electrocatalytic $CO_2$ reduction [45]. Therefore, the development of efficient electrochemical $CO_2$ reduction capable of promoting multielectron (and proton) transfer is a major task in this field.

### 3. Electrocatalysts

Over the past few decades, researchers have attempted to develop high-performance $CO_2$ electrocatalysts and have carried out extensive work [71–73]. The catalysts could reduce the activation energy required for the electroreduction of $CO_2$, thereby minishing the reduction overpotential and current density. Most of the studies have focused on transition-metal-based catalysts, which are mainly Cu, Au, Fe, Ag, Re, Mn, Co, Ni, Pd, Ir, Ru, etc. [17,74–76]. The main forms of the complexes include transition metal polypyridines, metal porphyrins/phthalocyanines, and various metal phosphine complexes [77,78]. It was found that transition metal complexes were used for the electrocatalytic reduction of $CO_2$ and presented outstanding performance. Some reported electrocatalysts for electrocatalytic $CO_2$ reduction are summarized in Table 2 and are illustrated in detail in the following sections.

**Table 2.** Summary of the performance of typical electrocatalysts for electrocatalytic $CO_2$ reduction.

| Electrocatalyst | Potential (V vs. RHE) | Major Products | FE (%) | Reference |
|---|---|---|---|---|
| Cu–In alloys (In: 80 at%) | −1.0 | formate | 62.0 | 2017 [79] |
| CuS@Ni Foam | −1.1 | methane | 73.0 | 2017 [80] |
| $Co(CO_3)_{0.5}(OH)\cdot0.11H_2O$ | −0.98 | methane | 97.0 | 2018 [81] |
| Co/Zn@ZIFs | −0.52 | CO | 94.0 | 2018 [82] |
| ultrathin Pd nanosheets | −0.5 | CO | 94.0 | 2018 [83] |
| Mn−doped $In_2S_3$ | −0.9 | formate | 86.0 | 2019 [84] |
| $Ni_1$–$N_2$–C | −0.8 | CO | 96.8 | 2019 [85] |
| Ni/Fe–N–C–DAC | −0.7 | CO | 99.0 | 2019 [86] |
| Pd–Au | −0.5 | CO | 80.0 | 2019 [87] |
| ultrathin porous Cu nanosheets | −1.0 | CO | 74.1 | 2019 [88] |
| Cu nanocubes | −0.5 | $C_2H_4$ | 60.0 | 2019 [89] |
| Fe–$N_5$–C | −0.46 | CO | 97.0 | 2019 [90] |
| Fe–$N_4$–C | −0.5 | CO | 94.9 | 2019 [91] |
| $Fe^{3+}$–N–C | −0.45 | CO | 90.0 | 2019 [92] |
| Ni−graphene oxide | −0.63 | CO | 96.5 | 2019 [93] |
| Ni−$N_4$−C | −0.65 | CO | 90.0 | 2019 [94] |
| Ni−$N_2$−C | −0.8 | CO | 98.0 | 2019 [95] |
| InN NSs | −0.9 | formate | 91.0 | 2020 [96] |
| NiSn−APC | −0.82 | formate | 86.1 | 2020 [97] |
| $Ni_{20}$−N−C | −0.53 | CO | 97 | 2020 [98] |
| Cu−Al | −1.50 | $C_2H_4$ | 80 | 2020 [99] |
| 5 nm $In_2O_3$ NPs | −0.7 | formate | 80.0 | 2021 [100] |
| $Pb_1Cu$ | −0.8 | formate | 96.0 | 2021 [101] |
| SAC−Ag/g−$C_3N_4$ | −0.7 | CO | 93.7 | 2021 [102] |
| 40Ni@N−C/rGO | −0.97 | CO | 92.0 | 2021 [103] |
| polycrystalline $SnS_x$ NFs | −1.0 | formate | 97.0 | 2022 [104] |
| $In_2O_3$@In−Co PBA | −0.96 | formate | 85.0 | 2022 [105] |
| Sb−SAs/NC | −0.8 | formate | 94.0 | 2022 [106] |
| Nb−N−C | −0.8 | CO | 90.0 | 2022 [107] |

Among the transition metal complexes, Cu complexes had better activity. Recently, Raja et al. [78] designed a dinuclear Cu (I) complex as catalysts for the electroreduction of $CO_2$, and its catalytic cycle is shown in Figure 2 below. They found that the Cu (I) system could be oxidized by $CO_2$, which suggested that the selective bonding of $CO_2$ and Cu (I) ions could provide a low-energy pathway for the formation of $CO_2^{\bullet-}$ radical anions. Thus, the Cu (II) tetranuclear oxalate-bridged complex $[2]^{4+}$ was thermodynamically favorable. Moreover, the bonding of $CO_2$ to the Cu(I) center in the dinuclear copper (I) complex $[1]^{2+}$ was faster and has a higher selectivity. This was due to the low solubility of lithium oxalate in acetonitrile, and the release of the oxalate double anion from $[2]^{4+}$ in the presence of $LiClO_4$ could be accomplished instantaneously, and the dinuclear copper (II) complex $[4]^{4+}$ was produced. Consequently, the electrocatalytic reduction of Cu (II) ions to Cu (I) might be the rate-controlling step in this system. The electrode surface was coated by the lithium oxalate generated during the reaction and then prevented the effective electron transfer.

Cu electrodes have good catalytic performance for the conversion of $CO_2$ to alkanes and alcohols [64]. The moderate hydrogen evolution overpotential of Cu electrodes can properly suppress the hydrogen production. Therefore, Cu electrodes can generate relatively high current efficiency [108]. Recently, Matthew et al. [109] prepared Cu-modified electrodes by calcining Cu sheets in air and then further electrochemically reducing and calcining the generated $Cu_2O$ (Figure 3). The activity exhibited by such electrodes in reducing $CO_2$ was highly dependent on the initial thickness of the $Cu_2O$ layer. The experimental results showed that the activity of the electrodes prepared from a $Cu_2O$ thin layer at 130 °C was not different from that of polycrystalline Cu. However, the $Cu_2O$ electrode formed by calcination at 500 °C with a thickness of not less than 3 μm had a large roughness coefficient, and its overpotential was 0.5 V smaller than that of polycrystalline Cu in the reduction of $CO_2$. More significantly, the current density of this electrode would be higher than

1 mA/cm$^2$ at an overpotential lower than 0.4 V, which was larger than the reported activity of other metal electrodes. Meanwhile, the Cu-modified electrode obtained by calcination at 500 °C remained stable after reacting for 7 h, while the polycrystalline Cu electrode started to passivate within 1 h under the same circumstances.

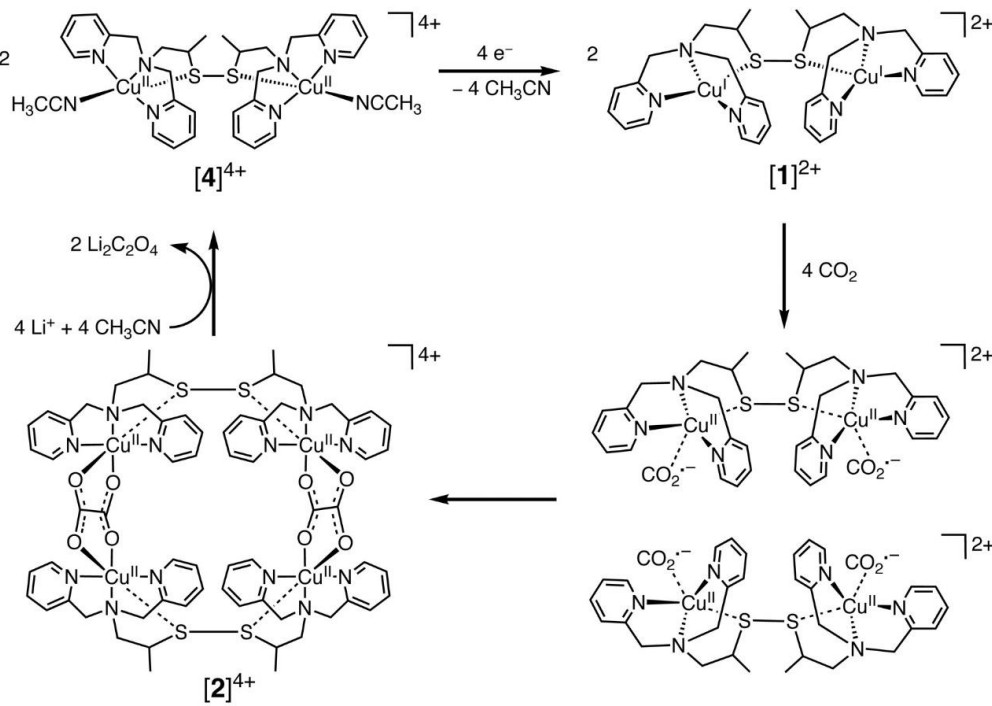

**Figure 2.** Proposed electrocatalytic cycle for oxalate formation. Reprinted from Ref. [78], copyright (2010), with permission from the American Association for the Advancement of Science.

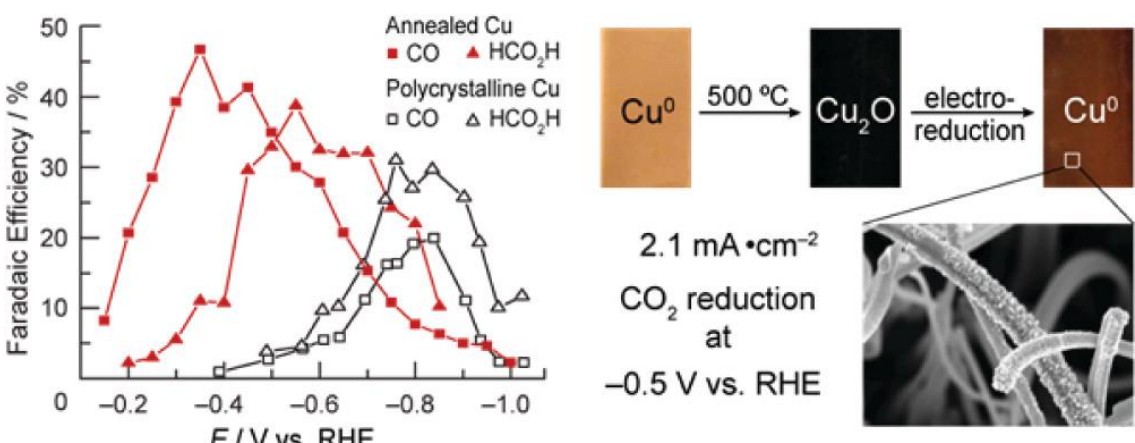

**Figure 3.** Electrocatalytic performance and the preparation process of Cu electrodes. Reprinted from Ref. [109], copyright (2012), with permission from the American Chemical Society.

To confirm that different components of binary nanocrystals have different effects on the performance of the electrocatalytic reduction of $CO_2$, Guo et al. [110] investigated the performance of Cu-Pt binary alloy nanocrystals for the electrochemical reduction of $CO_2$ in 0.5 M $KHCO_3$ at room temperature. As shown in Figure 4, it was found that among Cu-Pt nanocrystals with different molar ratios, the catalyst with a molar ratio of 3:1 had the best activity for the catalytic reduction of $CO_2$, the lowest onset potential ($-0.972$ V), and the highest current density (0.598 mA/cm$^{-2}$, $-1.3$ V vs. SCE). The optimization of catalyst components was compared, and a reasonable mechanism hypothesis was proposed. As illustrated in Figure 5, Pt had a unique adsorption capacity for protons, and it was

an efficient catalyst for HER, leading that $H_2O$ molecules in solution would be adsorbed on Pt atoms in Cu-Pt nanocrystals. CO*, the reaction intermediate generated by $CO_2$ gaining electrons, was adsorbed on Cu atoms, and then combined with $H^+$ provided by $H_2O$ to generate HCO* and finally $CH_4$. The above mechanism well explains why Cu-Pt nanocrystals with a molar ratio of 3:1 had the highest activity in the catalytic reduction of $CO_2$. In the process of $CH_4$ generation, CO* and $H^+$ were indispensable, and Cu promoted $CO_2$ to gain electrons to generate CO*. Pt facilitated the adsorption of $H_2O$ to generate $H^+$, so the molar ratio of Cu-Pt had an optimal value; too much Pt or Cu was not conducive to $CH_4$ production.

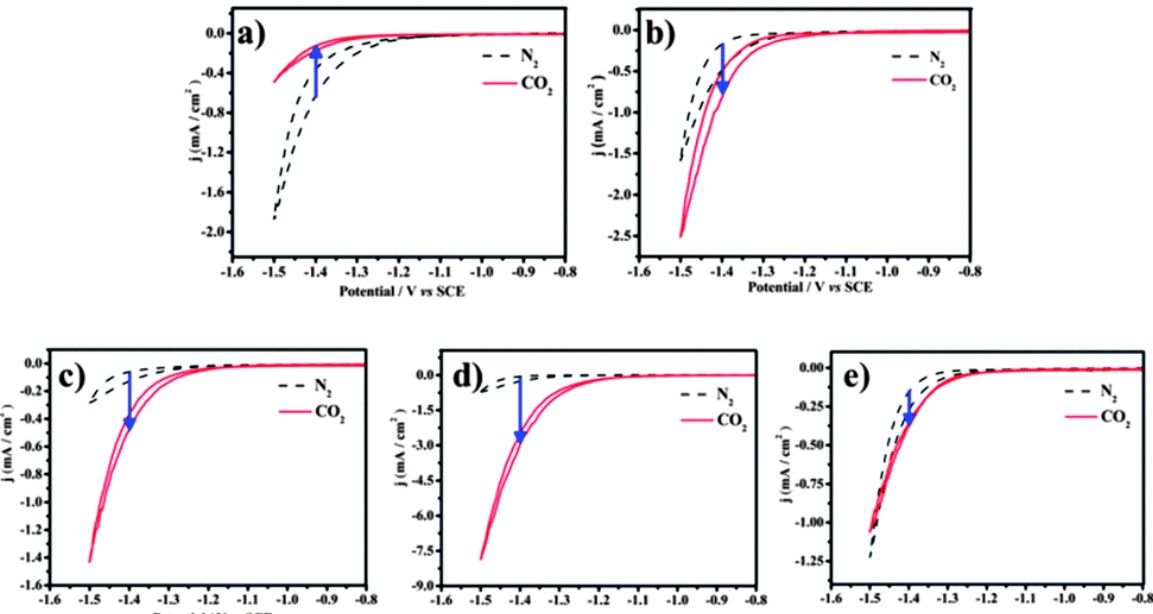

**Figure 4.** The cyclic voltammograms of catalysts were recorded in $N_2$- and $CO_2$-saturated 0.5 M $KHCO_3$ with a scan rate of 10 mV/s$^{-1}$ between $-0.8$ and $-1.5$ V (vs. SCE). (**a**) Cu-Pt-1#, (**b**) Cu-Pt-2#, (**c**) Cu-Pt-3#, (**d**) Cu-Pt-4#, and (**e**) Cu-Pt-5#. Reprinted from Ref. [110], copyright (2015), with permission from the Royal Society of Chemistry.

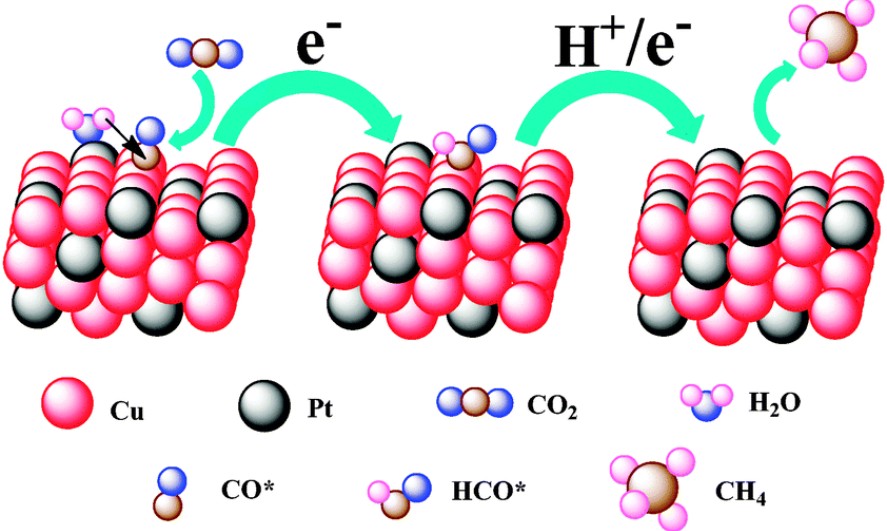

**Figure 5.** A proposed mechanism illustrating the steps of $CO_2$ electroreduction and $CH_4$ formation occurring at the Cu-Pt 3:1 NC catalyst. Reprinted from Ref. [110], copyright (2015), with permission from the Royal Society of Chemistry.

However, Cu-based catalysts still faced some problems because the instability of Cu materials was easily deactivated and decomposed during the catalytic reduction of $CO_2$. Some prepared Cu-based catalysts were easily oxidized when exposed to air [21,75,109]. The valence and morphology changes of Cu-based catalysts during the catalytic process should be deeply elucidated, which conduced to provide a crucial role. Ni et al. [111] proposed a modulation strategy: hydrogen reduction valence, which could simply regulate the ratio of different valence states of Cu by optimizing the reduction time, and helped to investigate the mechanism of multivalent Cu in depth. The experimental results showed that the excellent performance of G-Cu$_x$O-2h not only stabilized the intermediate product $CO_2^{\bullet-}$, but also accelerated the rate-limiting step in the $HCOO^-$ desorption process, which was related to the optimal Cu (I) content in the catalyst. The paper also proposed a "buffering effect" to explain the stability of G-Cu$_x$O-2h, as shown in Figure 6. Cu (II) from the thicker subsurface layer acted as a sacrificial source to supplement Cu (I), thus balancing the Cu (I) content in the surface layer and maintaining the activity of the catalyst in the reaction.

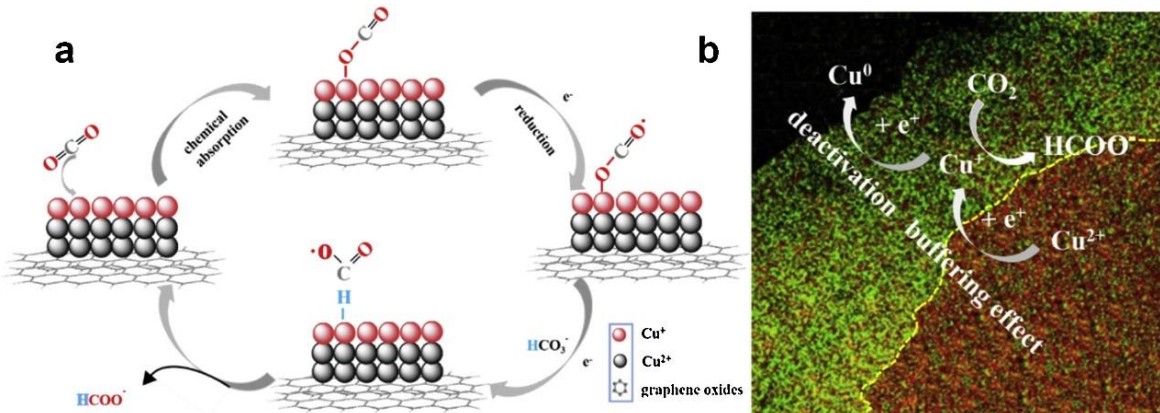

**Figure 6.** (**a**) A proposed mechanism illustrating the steps of electrochemical $CO_2$ reduction and the formation of HCOOH occurring at the G-Cu$_x$O-2 h electrocatalyst. (**b**) A proposed "buffering effect" illustrating the origin of the durability of G-Cu$_x$O-T electrocatalysts. Reprinted from Ref. [111], copyright (2019), with permission from Elsevier.

Furthermore, the electrocatalytic reduction of $CO_2$ at the Au electrode has also been investigated by Matthew et al. [112]. They used the periodically symmetric square wave potential method to prepare an amorphous $Au_2O_3$ layer on a Au electrode. This electrode was used directly for the electrocatalytic reduction of $CO_2$, in which $Au_2O_3$ was reduced to Au within 15 min. The electrode exhibited high selectivity for the product CO in the reduction of $CO_2$ with an overpotential of only 0.14 V, and the activity was maintained for at least 8 h. The authors attributed the high activity of such catalysts to the increased stability of the $CO_2^{\bullet-}$ intermediate; the electrolyte $HCO_3^-$ acted as $H^+$ donors during the catalytic process (Figure 7).

Gao et al. synthesized a Pd nanoparticle electrode and efficiently catalyzed the reduction of $CO_2$ to CO. They found that Pd nanoparticle size exhibited significant size dependence in the range of 2.4–10.3 nm [113]. With the Pb nanoparticle size changes from 10.3 to 3.7 nm, the Faraday conversion efficiency of CO generation at −0.89 V increased from 5.8% to 91.2%, while the current density of CO generation was enhanced 18.4 times (Figure 8). The relationship between catalyst performance and particle size was obtained using the density functional theory (DFT) to further analyze the free energy of $CO_2$ reduction and HER at three different reaction sites: plane, step, and corner. The results indicated that the relationship between the conversion frequency (TOF) of CO generation and catalyst particle size presented a volcano-like curve (Figure 9). This illustrated that $CO_2$ adsorption, COOH* formation, and CO* removal could be modulated by changing

the size of Pb nanoparticles, thus enabling the transition of Pd nanoparticles from HER catalysts to efficient $CO_2$ reduction catalysts.

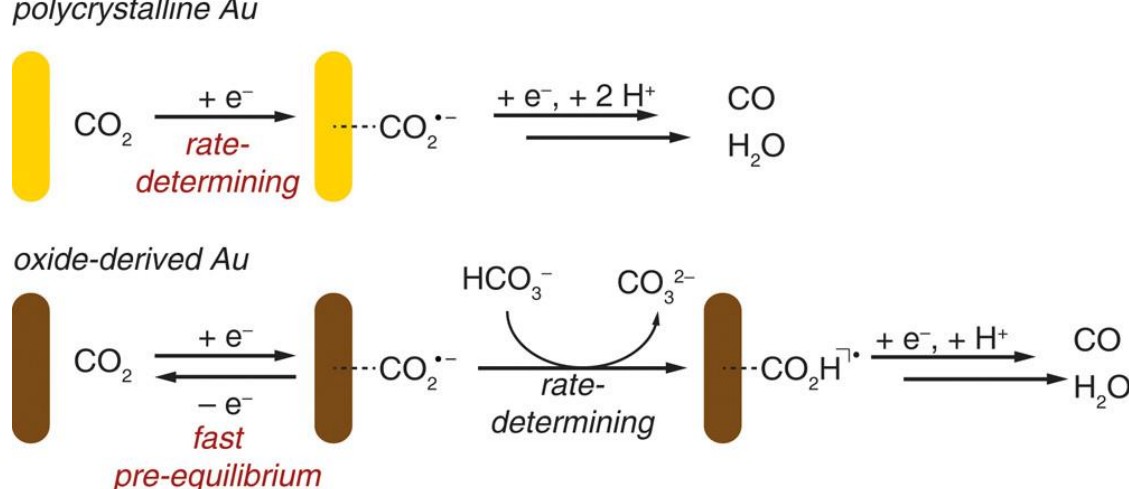

**Figure 7.** Proposed mechanisms for $CO_2$ reduction to CO on polycrystalline Au and oxide-derived Au. Reprinted from Ref. [112], copyright (2012), with permission from the American Chemical Society.

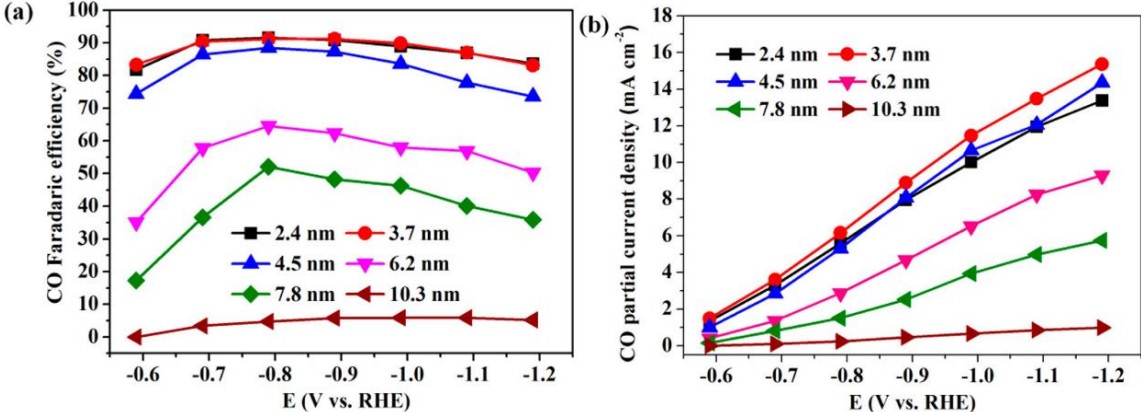

**Figure 8.** Applied potential dependence of (**a**) Faradaic efficiencies and (**b**) current densities for CO production over Pd NPs with different sizes. Reprinted from Ref. [113], copyright (2015), with permission from the American Chemical Society.

As well known, the metal electrode possessed high activity for the catalytic reduction of $CO_2$ [114,115]. The outstanding catalytic activity was obtained by reducing metal oxides to metal electrode [116–118]. This electrode could decrease the reduction potential of $CO_2$ to a thermodynamic minimum. In comparison, other preparation methods caused many microstructural changes in the catalysts, such as interfaces and defects [119–121]. The influence of metal oxides on the catalytic performance of metals was related to these microstructures. However, the mechanism of the above system was not clear [122,123]. Recently, a 4-atomic-layer-thick Co oxidized heterostructure (see Figure 10) was prepared by Gao et al. by means of ligand-confined growth and used to explain the effect of metal surface oxides on their own electroreduction $CO_2$ properties [124]. This work demonstrated that metal atoms located in specific oxidation states enabled the production of higher catalytic activity through specific arrangements, thus providing a new idea for the development of efficient and stable catalysts for $CO_2$ reduction. It was important for promoting research on the mechanism of metal oxides' electrocatalytic reduction of $CO_2$.

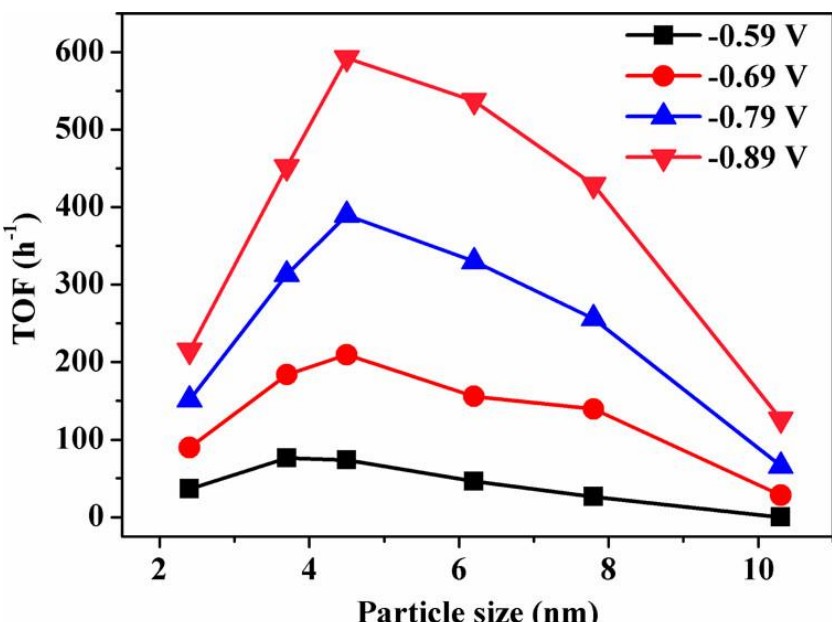

**Figure 9.** Size dependence of TOF for CO production on Pd NPs at various potentials. Reprinted from Ref. [113], copyright (2015), with permission from the American Chemical Society.

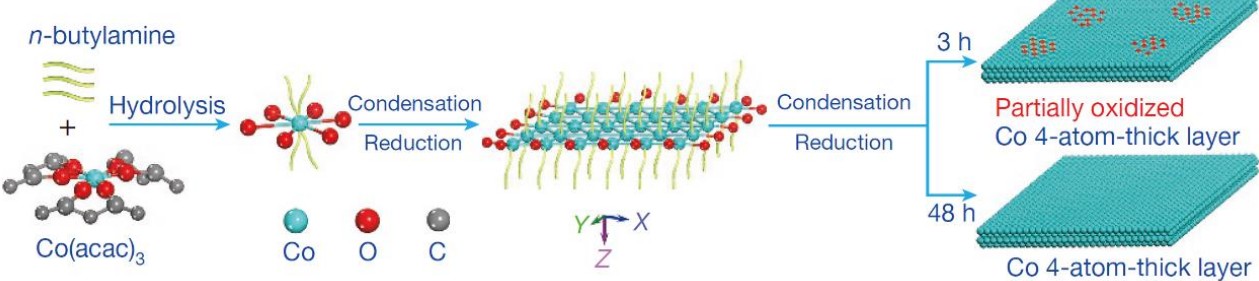

**Figure 10.** Schematic formation process of the partially oxidized and pure-Co 4-atomic-layer, respectively. Reprinted from Ref. [124], copyright (2016), with permission from Springer Nature.

Transition metal porphyrins have also shown superior reactivity in catalysts for the electrocatalytic reduction of $CO_2$, such as iron porphyrins and cobalt porphyrins [125]. Cyrille et al. found that the modified iron porphyrins accelerated the reduction reaction of $CO_2$ after the introduction of phenolic groups in all the neighbors of the iron porphyrin phenyl group. It was concluded that the increased activity of iron porphyrins was related to the high local concentration of protons associated with the phenolic hydroxyl substituents; the relationship between the high turnover frequency (TOF) and the small overpotential (η) of the catalyst was systematically investigated (Figure 11) [77].

Cobalt porphyrins as catalysts for the electrocatalytic reduction of $CO_2$ have a relatively high overpotential (−1.3∼−1.6 V vs. SHE) and a low TOF. The possible catalytic mechanism of cobalt porphyrins was analyzed by Kevin et al. using DFT calculations [126]. From the calculations, it was clear that $CO_2$ was bonded to the particle $[Co(I)P]^-$, and the mechanism is shown in Figure 12. Besides, the presence of water played a crucial role because the key intermediates $[Co(I)P-CO_2]^{2-}$ and $[Co(II)P-CO_2H]^-$ were stabilized by hydrogen bonding interaction, which caused the exothermic breakage of the C-O bond. Theoretical calculations also revealed that the electron transfer between the gas diffusion electrode and the polymerized porphyrin catalyst was the rate-controlling step of the whole reaction. These findings were important implications for the capture and the electrochemical reduction of $CO_2$, respectively.

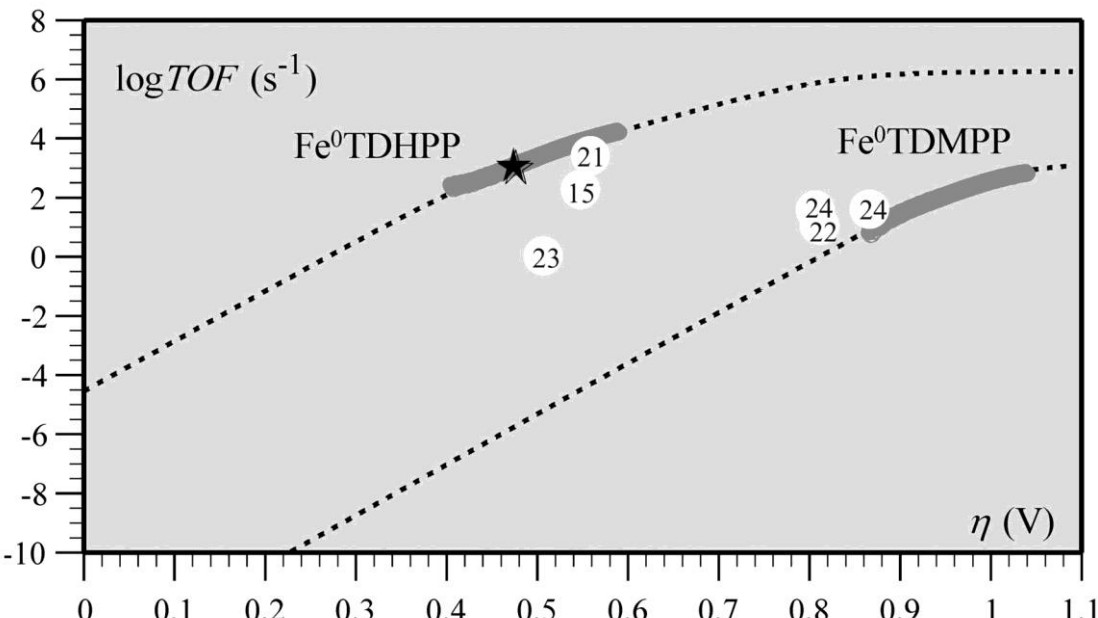

**Figure 11.** Correlation between turnover frequency and overpotential for the series of $CO_2$-to-CO electroreduction catalysts. The star indicates TOF and η values from preparative-scale experiments of $Fe^0$TDHPP. Reprinted from Ref. [77], copyright (2012), with permission from the American Association for the Advancement of Science.

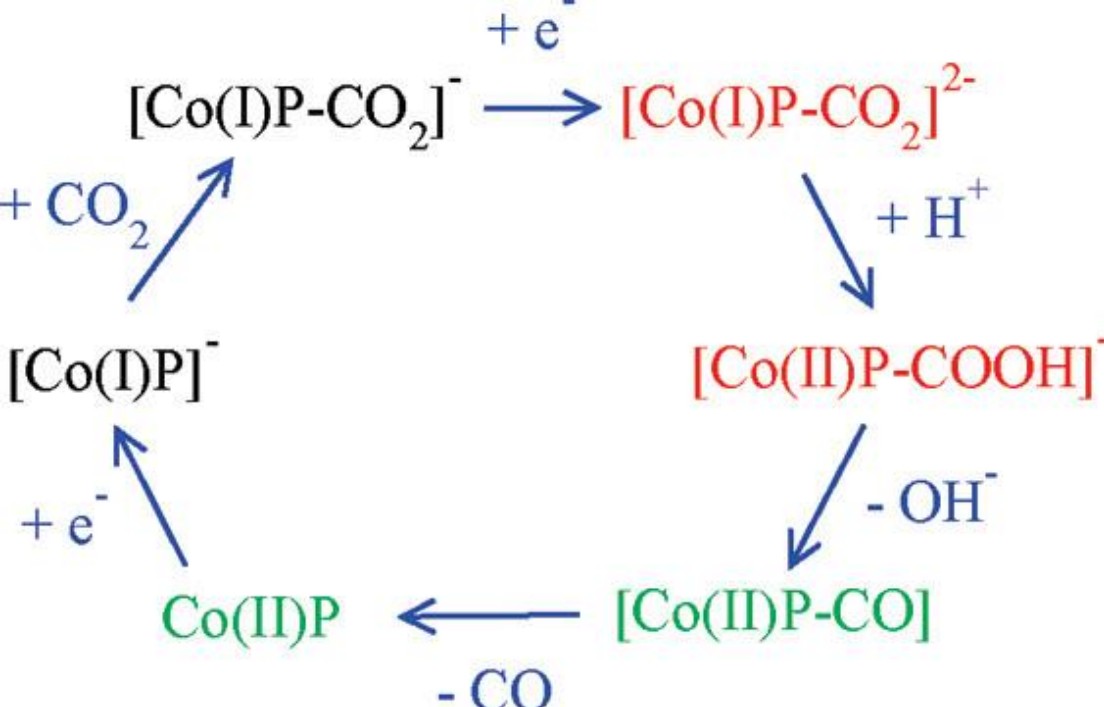

**Figure 12.** Mechanism of $CO_2$ reduction with electron addition deduced from hybrid DFT plus dielectric continuum redox potential calculations. Red denotes key intermediates; green species should undergo fast reactions. Reprinted from Ref. [126], copyright (2010), with permission from the American Chemical Society.

The effect of organometallic Ag catalysts supported by carbon–nitrogen in the electrochemical conversion of $CO_2$ was investigated by Claire et al. [127]. It was shown that such catalysts decreased the reaction overpotential and improved the selectivity, which also

led to an enhanced reaction rate for $CO_2$ reduction. Moreover, these nitrogen-containing compounds might act as cocatalysts in the electrochemical reduction of $CO_2$, create a coordinated effect with the Ag surface, and thus facilitate electron transfer. Remarkably, they found that not all nitrogen-containing atoms introduced to the carbon surface exhibited the properties of such catalysts. Therefore, to further enhance the activity and selectivity, a more in-depth study of the $CO_2$ reduction mechanism was needed to elucidate the interactions between these complexes and the carbon substrate.

Agarwal's group carried out an in-depth study of the $CO_2$ reduction mechanism. Although the results were based on the photocatalytic reduction of $CO_2$, their systematic analysis of the $CO_2$ reduction mechanism was equally applicable in the direction of electrocatalytic reduction of $CO_2$ [128]. First, they found that tricarbonyl Re complexes, such as Re(bpy)(CO)$_3$Cl (bpy=2,2'-dipyridine), exhibited high activity in catalytic $CO_2$ reduction in the presence of electron sacrificial agents. Subsequently, they investigated the potential pathway of formate generation by the Re-hydride insertion theory in the presence of triethylamine (TEA) using DFT. It was suggested that TEA was the main donor of hydrogen atoms and electrons, and its catalytic cycle pathway is shown in Figure 13.

**Figure 13.** Computed photocatalytic cycle for $CO_2$ reduction on a Re catalyst. Reprinted from Ref. [128], copyright (2011), with permission from the American Chemical Society.

## 4. Electrolyte

The influence of the composition of the electrolyte on the $CO_2$ reduction cannot be ignored [129,130]. Therefore, the study of electrolyte is of great importance to optimize the conditions for the electrochemical reduction of $CO_2$. Researchers have conducted numerous studies on the electrolyte used for the electrocatalytic reduction of $CO_2$ [131–133]. Agoritsa et al. found that the rate of electrochemical reduction $CO_2$ increased in the order of electrolyte cations: $Na^+ < Mg^{2+} < Ca^{2+} < Ba^{2+} < Al^{3+} < Zr^{4+} < Nd^{3+} < La^{3+}$, where the rate of $La^{3+}$ was twice as high as that of $Na^+$ at the same potential [134]. The increasing order of halogen anions was $Cl^- < Br^- < I^-$. In addition, the conclusions reached by different researchers about the effect of electrolyte ions often differed or even conflicted with each other, which may be due to the fact that researchers only discussed the effect of certain factors on the reaction process and neglected other factors, such as the operation time, the conductivity of the solution, the solubility of $CO_2$, the concentration of the product at the cathode, and some hydrodynamic factors and their interactions [135].

Moreover, the electrolyte synergistically promoted electrode catalytic reactions to accelerate the electrochemical reduction of $CO_2$ [136–140]. Kotaro et al. investigated the



effect of Cu wire electrodes in 3 mol L$^{-1}$ of KCl, KBr, and KI electrolytes with X$^-$ (Cl$^-$, Br$^-$, I$^-$) on the electrocatalytic reduction process of CO$_2$ [141]. It was shown that Cu-X acted as a catalytic layer, facilitating the transfer of electrons from the electrode to CO$_2$. The electron transfer to CO$_2$ might be accomplished by X–C bonding, and the X–C bonding was formed by electron flow between the specific adsorbed halogen anion and the empty orbital of CO$_2$ (Figure 14). The stronger the adsorption of the halogen anion on the electrode, the more CO$_2$ was bound, which in turn generated a higher CO$_2$ reduction current. The specific adsorbed halogen anion could also inhibit the adsorption of protons, which in turn generated a higher hydrogen overpotential. This interaction and influence minished the CO$_2$ reduction overpotential while allowing the rate of electrochemical CO$_2$ reduction to be enhanced.

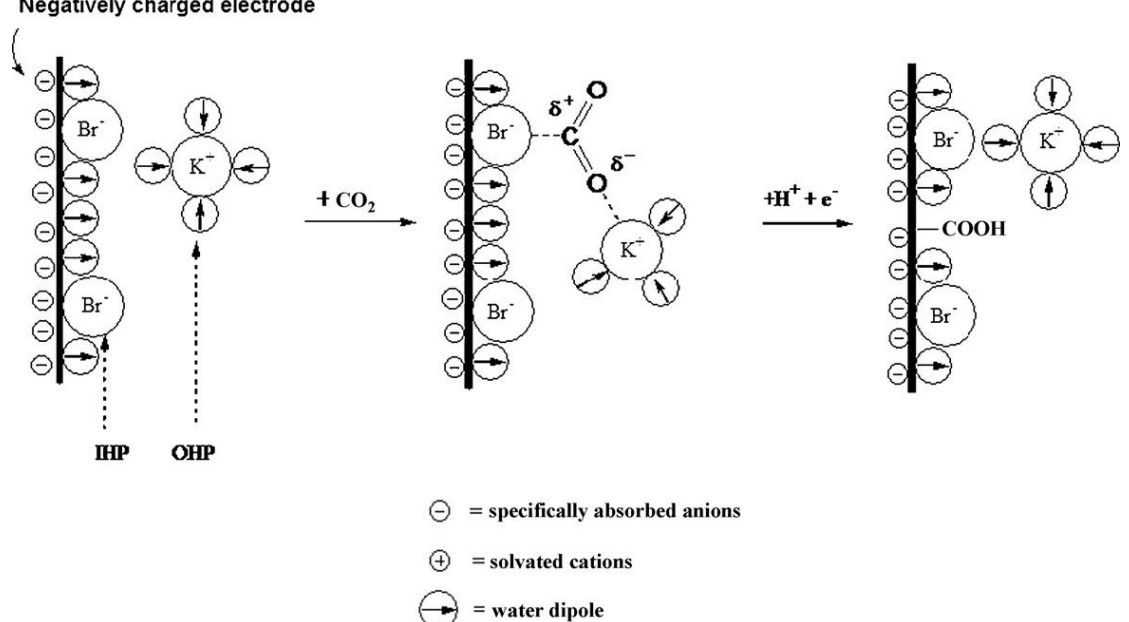

**Figure 14.** Schematic of specifically adsorbed anions (Br$^-$), approach of CO$_2$, and subsequent electrochemical reduction. Reprinted from Ref. [141], copyright (2010), with permission from Elsevier.

Nonaqueous organic solvents can also be used as electrolytes for the electrochemical reduction of CO$_2$ (such as CH$_3$CN) [142–144]. Electrocatalytic reduction of CO$_2$ in organic solvents has the following advantages over water: (1) the solubility of CO$_2$ is greater than in water; (2) competitive reactions for CO$_2$ reduction (H$_2$ generation) can be suppressed; (3) it is used as a potent and better CO$_2$ absorber for industrial applications, and the process is more energy efficient; and (4) the possibility of reduction below 0 °C is realized. Electrolytes are currently studied in several directions: methanol, ionic liquids, acetonitrile–ionic liquid mixtures, iodomethane–ionic liquid mixtures, and methanol–potassium–ionic liquid mixtures [145].

## 5. Reactor Design

The structural design of the reactor and the transport of the material have a greater influence on the CO$_2$ reduction [146,147]. In recent years, several articles have reported several reactor designs, most of which were based on fuel cell designs and used polymer electrolyte membranes to separate the anode from the cathode. Subramania used a composite perfluoropolymer cation exchange membrane (Nafion) to separate the anode from the cathode and to perform the CO$_2$ reduction reaction at room temperature [148]. This continuous reactor was a great improvement over the intermittent reactor. The maximum current efficiency of formate formation reached 93%; the concentration of formate was up to 0.015 mol L$^{-1}$.

The alkaline polymer electrolyte membrane cell for the electrochemical conversion of $CO_2$ was further investigated by Narayanan et al. [149]. The specific structure of this cell is shown in Figure 15. The advantages of this type of reactor were that (1) the reduction products of $CO_2$ could be saved from reoxidation by the $O_2$ electrode, (2) a nonprecious metal and its oxide could be used as catalysts, (3) the integrated compact porous electrode structure achieved a low internal resistance of the cell, and (4) it was scaled up to large sizes with no efficiency loss.

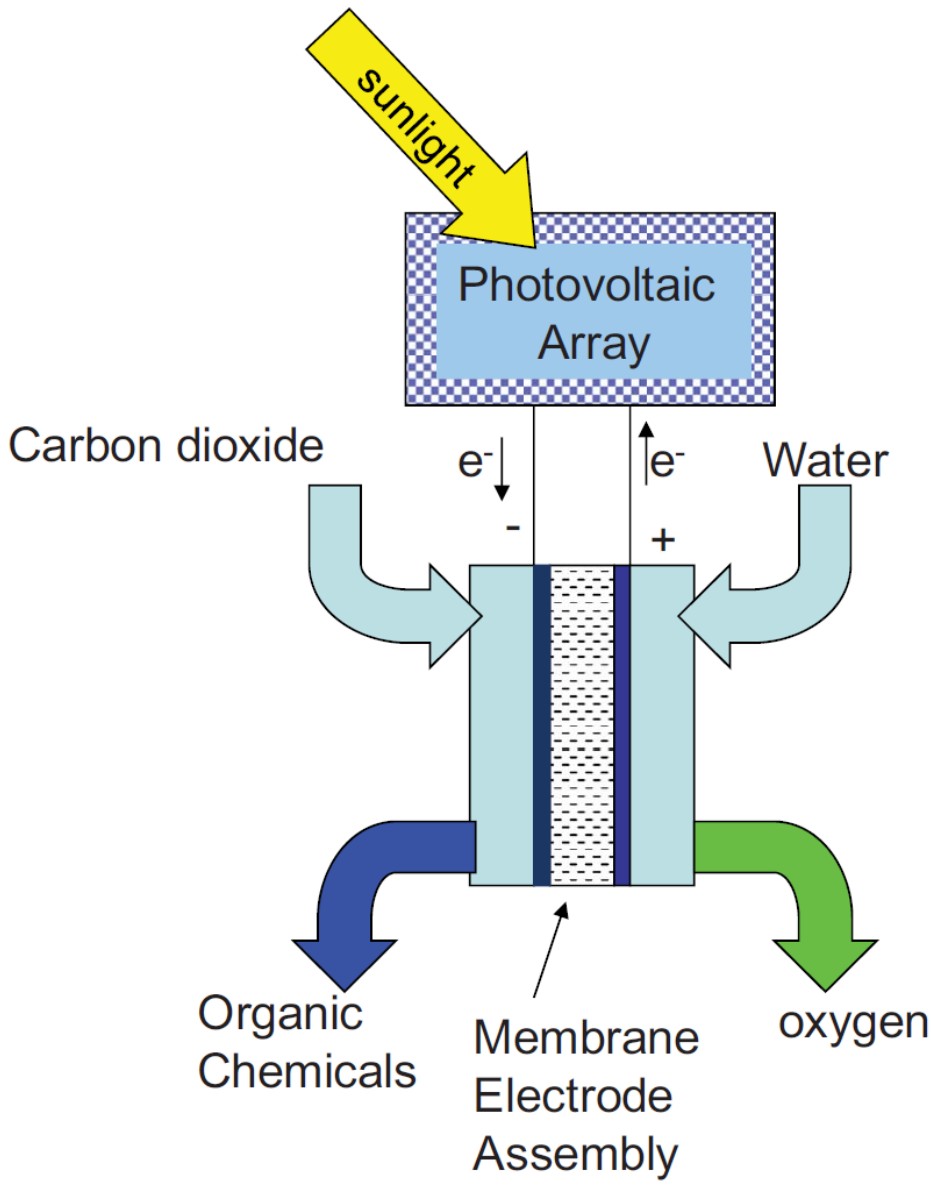

**Figure 15.** Polymer membrane cell configuration for the electrochemical reduction of carbon dioxide. Reprinted from Ref. [149], copyright (2010), with permission from the Electrochemical Society.

Besides, Claudio et al. designed a novel photoelectrocatalytic (PEC) reactor for the synthesis of solar fuels [150]. The internal configuration of this device is depicted in detail in Figure 16. The cathode was made by depositing a suspension of Fe or Pt carbon nanotubes (CNT) on carbon cloth with ethanol, and the cathode was applied to reduced $CO_2$ to liquid fuel (isopropanol as the main product) in the gas phase. The simplified process of this reactor was: (1) light passed through the quartz window to reach the photoanode, which in turn generated photogenerated electrons and holes to produce $O_2$; (2) protons passed through the Nafion membrane, and electrons were collected through the external wire

to reach the cathode; and (3) under the action of the CNT electrocatalyst, electrons and protons reacted with $CO_2$ to produce liquid fuel, or alternatively, protons and electrons reacted on carbon cloth-supported Pt nanoparticles to produce $H_2$.

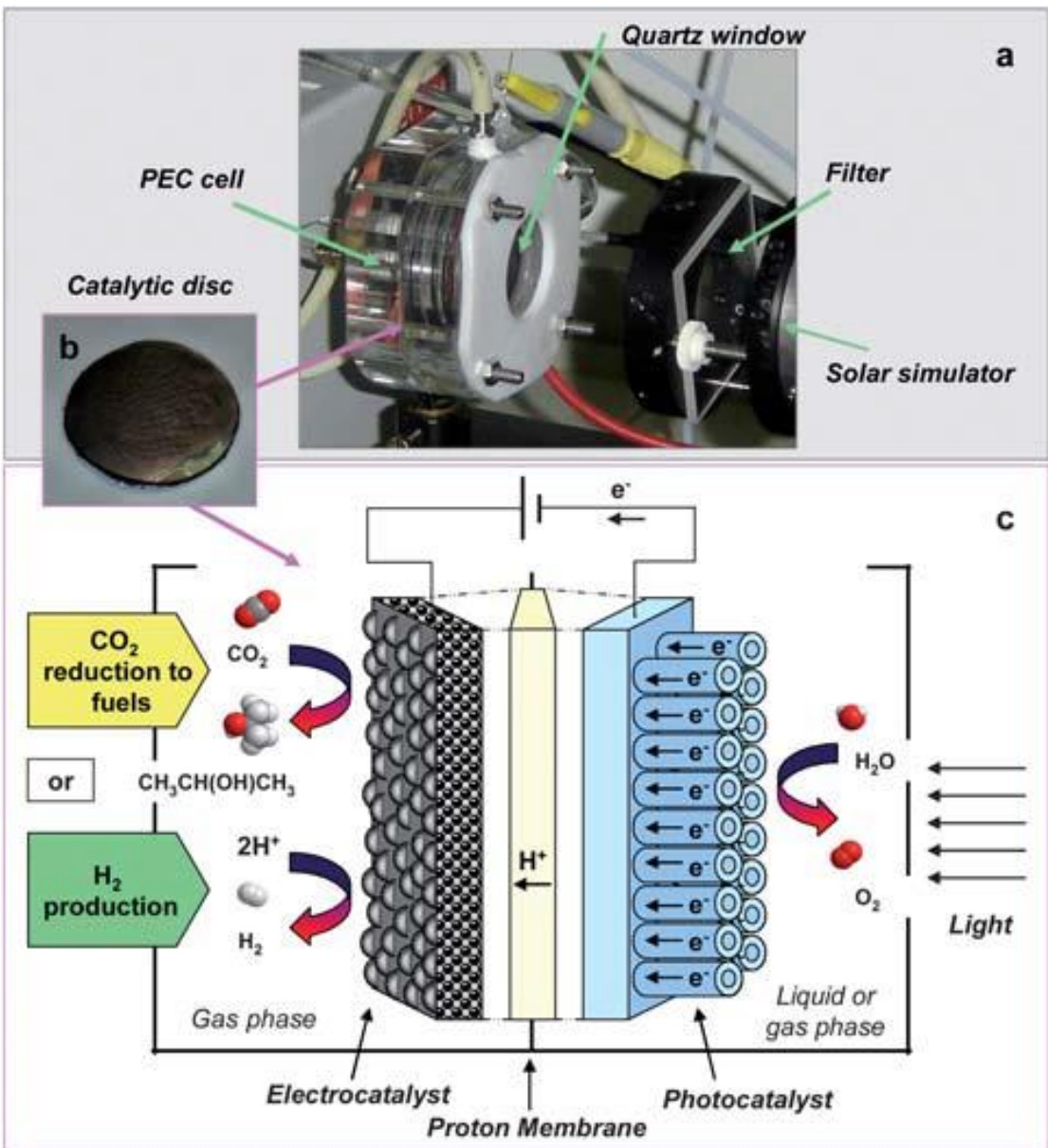

**Figure 16.** (**a**) View of the lab-scale PEC device. (**b**) Image of the photo/electrocatalytic disc. (**c**) Scheme of the PEC device for $CO_2$ reduction to fuels and $H_2$ production. Reprinted from Ref. [150], copyright (2010), with permission from the Royal Society of Chemistry.

Devin et al. reported a microfluidic reactor [151]. It has a structure similar to that of the microfluidic $H_2$/$O_2$ fuel cell reported by the group. The cathode and anode of this reactor were separated by a flowing liquid electrolyte [152], and its structure is schematically shown in Figure 17. The study demonstrated that the microfluidic electrochemical cell could be applied as an effective reactor and a versatile analytical tool for the electrochemical reduction of $CO_2$. The novelty of the design lay in the flowing liquid electrolyte stream, the advantages of which were mainly in the following aspects: (1) the wide flexibility of the working environment, especially in terms of electrolyte composition and pH; (2) the flow of the electrolyte to the anode provided one of the reactants, $H_2O$, for the reaction

to proceed, while reducing the problem of water management on the electrode surface; (3) the continuous flowing environment facilitated the online collection of samples and allowed for fast and simple analysis of the products; and (4) a reference electrode was placed at the exit stream to promote the analysis of the performance of each electrode. In addition, this cell had a high efficiency (89% response current efficiency and 45% energy efficiency), and its current density reached 100 mA cm$^{-2}$. Jaramillo et al. designed a $CO_2$ electroreduction microreactor [153], as shown in Figure 18. The cathode and anode of the reactor were separated by an anion exchange membrane to prevent the interaction of the liquid-phase products of the two poles, and the exit gas was directly passed to the chromatograph for analysis.

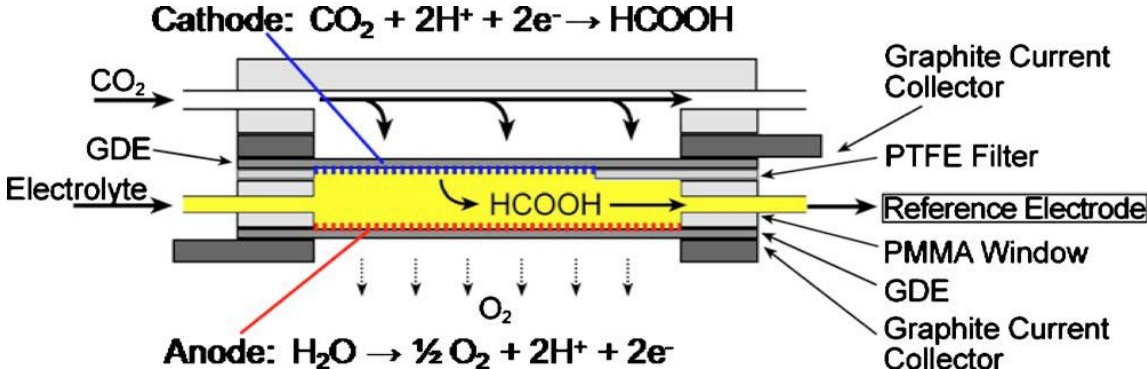

**Figure 17.** Schematic diagram of the microfluidic reactor for $CO_2$ conversion. Reprinted from Ref. [151], copyright (2010), with permission from the Electrochemical Society.

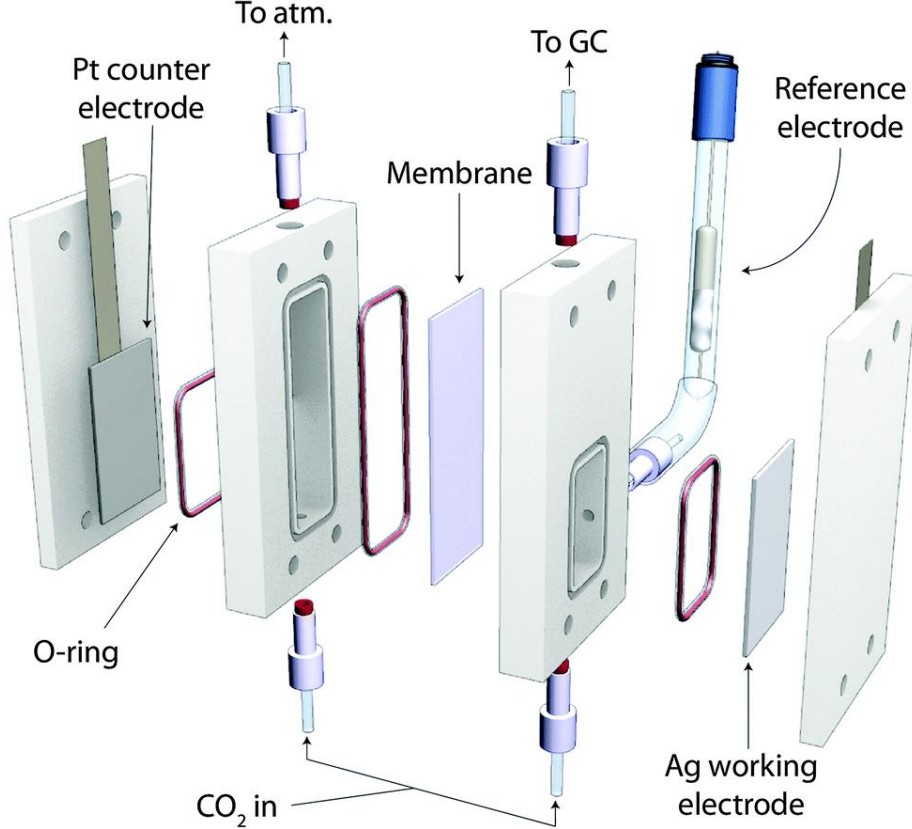

**Figure 18.** Schematic of the electrochemical cell utilized in this work. Reprinted from Ref. [153], copyright (2014), with permission from the Royal Society of Chemistry.

## 6. Conclusions and Outlook

The study of electrocatalytic $CO_2$ reduction is of great importance for spacecraft life support systems and energy storage conversion. However, research on $CO_2$ electrocatalytic reduction is still at the laboratory stage due to the high overpotential of $CO_2$ electrocatalytic reduction, the low reduction yield, and the lack of in-depth research on the catalytic mechanism. In order to solve these problems, researchers need to make continuous improvements in catalyst, electrolyte, and reactor design, and explore efficient methods for the electrocatalytic reduction of $CO_2$.

(1) The core of electrocatalytic $CO_2$ reduction is how to prepare efficient electrocatalysts that can use their catalytic activity to reduce external electron energy input and energy consumption, while improving the selectivity and controllability of the reduction products. Meanwhile, the catalyst must be able to achieve multielectron and multiproton transfer to enable efficient electrocatalytic reduction of $CO_2$ on the same surface.

(2) It is necessary to find new electrolyte systems that can synergize with the catalyst in organic systems and to analyze the specific functions of the electrolyte in the electrocatalytic reduction of $CO_2$ and the mechanism of action in situ in order to better understand the electrocatalytic reduction of $CO_2$.

(3) Although the design of the reactor is still at a preliminary stage, the size, shape, and structure of the reactor are of great importance to improve the efficiency and selectivity of electrocatalytic $CO_2$. Therefore, to further promote the practical application of $CO_2$ electrocatalytic reduction, this research needs to be further strengthened.

In conclusion, as an effective means of $CO_2$ recycling in space stations, the electrocatalytic reduction of $CO_2$ has bright research prospects, and this field can further enrich catalytic science and catalytic technology, thus advancing the progress of scientific research in related fields.

**Author Contributions:** Conceptualization, C.R. and W.N.; supervision and resources, H.L.; data curation and visualization, W.N.; writing—original draft preparation, C.R.; writing—review and editing, C.R.; funding acquisition, H.L. All authors have read and agreed to the published version of the manuscript.

**Funding:** This work was funded by the Natural Science Foundation of Guangxi (Nos. 2021GXNS-FAA220108 and 2020GXNSFBA297122) and National Key Research and Development Program (No. 2022YFEO134600).

**Data Availability Statement:** Data are available in the manuscript.

**Conflicts of Interest:** The authors declare no conflict of interest.

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
