# Peer review of "Recent Progress in Electrocatalytic Reduction of CO2"

_catalysts, doi:10.3390/catal13040644_

Round 1

Reviewer 1 Report

In their manuscript, Ren et al. provide an overview of recent progress in the electrocatalytic reduction of CO2. The authors begin with a broad introduction that highlights the importance of reducing CO2 emissions in space stations, and discuss various strategies for achieving this goal. While, this is indeed one of the application for CO2 fixation, this is not the only purpose, and shouldn’t be the main theme of the introduction part. Then, the authors presented a very general description of the catalytic mechanism of CO2 reduction reaction (CO2RR). Further, the discussion of the electrocatalysts investigated in CO2RR lacks coherence. The examples are not logically selected based on catalyst type (homogeneous versus heterogeneous), product type, or catalytic material nature. The manuscript concludes with a discussion of points related to electrolyte and reactor design. However, overall, the manuscript lacks organization structure, and in-depth discussions, and would benefit from further revisions before being submitted for publication in a scientific journal such as "Catalysts."

Reviewer 2 Report

This review article by Prof. Li et al. summarizes recent progress of electrocatalytic CO2 reduction. The coverage includes the heterogeneous electrode reaction and homogeneous reaction of CO2 electro-reduction, the influence of electrolytes, reactor designs. Given a vast number of recent developments of related studies, a short review is probably not the best format to report this important topic. This Reviewer understands the efforts of the author, trying to cover a broader scope on the title topic, which is unfortunately not a good idea. There are a few comments to the review article.

(1) The author has emphasized the importance of CO2 reduction reactions with respect to a stable life support in spacecrafts by controlling the CO2 concentration via electrochemical reactions of CO2. It will be much clearer to readers when the key factors (or differences) for CO2 reduction reactions to be conducted in space are stated in the introduction. Without technical requirements, there is no point to mention the spacecraft issue.  

(2) As stated early, the coverage of the review article is too large. It is recommended to limit to some specific areas. There are so many literature results to date. A lot of important results are not discussed in the article. To avoid this kind of fouls, the author may want to focus on heterogeneous reactions (Cu, Au, Pt electrodes, etc), organometallic/coordination complex-involved reductions, electrolyte influences, reactor design for optimal performance, or key factors concerning selectivity. It will give a much better impression. 

(3) The other way to present the review is: only present the best results in literatures. Discuss the difference between the best and the rest. Any important finding, key issues for the superior performance.

(4) It is recommended to arrange all reactions in the ascendent order of # of electrons involved. 

(5) L90-92, catalysts can reduce the activation energy while it is not necessary to increase the reaction selectivity. 

(6) Line 104, define [1]2+.

Reviewer 3 Report

Dear C. Ren et. al. your manuscript is highly relevant and I only have two questions:

Comment #0: I feel that the CO2 reduction mechanisms are missing in a more detailed way.

Comment #02: A graph is missing with the manuscripts published by year, the last 5 years.

Round 2

Reviewer 1 Report

I think the revised Manuscript, though didn't have substantial changes from the previous version, it can meet the criteria for publication in "Catalysts".

Reviewer 2 Report

The revision is good for publication. This Reviewer recommends the acceptance of the manuscript to the Journal.